# INVARIANT AND EQUIVARIANT ARCHITECTURES VIA LEARNED POLARIZATION

## ABSTRACT

We present a theoretical framework for constructing invariant and equivariant neural network architectures based on polarization methods from classical invariant theory. Existing approaches to enforcing symmetries in machine learning models often rely on explicit knowledge of the invariant ring of a group action, which is computationally demanding or intractable for many groups. Our framework leverages polarization to generate separating sets of invariant polynomials on high-dimensional group representations from those of lower-dimensional ones. We establish conditions under which separating sets can be obtained via standard, simple, or cheap polarization and demonstrate how these results can be combined with recent advances on separating families to yield small, expressive sets of invariants. This construction ensures universal approximation of continuous invariant functions while reducing computational complexity. We further discuss the implications for designing scalable invariant and equivariant architectures and identify settings where polarization provides a practical advantage, particularly for high-dimensional representations of finite groups.

## 1 INTRODUCTION

Enforcing symmetries in neural network architectures has been shown to improve their performance, generalization capabilities, and data efficiency across various applications. These symmetries are in general expressed as invariance or equivariance of the network with respect to the action of a group on the input data. Methods to construct invariant and equivariant architectures include the use group convolutions (Cohen & Welling, 2016; Cohen et al., 2018) or the use of explicit invariant and equivariant functions (Zaheer et al., 2017; Maron et al., 2018; Kondor & Trivedi, 2018; nathaniel thomas et al., 2018; Satorras et al., 2021; Simeon & De Fabritiis, 2023; Deng et al., 2021). Although these methods have been successfully applied to a variety of problems, they often rely on the specific knowledge of the group action and the resulting ring of invariant polynomials it induces. While these rings are well characterized for some common groups, such as the symmetric group or the orthogonal group, they can be difficult to determine for other groups. In this paper, we introduce a theoretical framework based on polarization methods from invariant theory to construct invariant and equivariant architectures for high-dimensional representations of a group from low-dimensional ones. This framework is applicable when the high-dimensional representation can be decomposed as a direct sum of smaller representations on which the group acts diagonally. We show that under certain conditions, a set of polynomials that separates orbits in the high-dimensional representation can be obtained by polarization from a separating set of polynomials on a smaller representation. Since separating sets are known to be easier to compute and smaller in size than generating sets, this result provides a practical way to construct invariant functions that are guaranteed to be sufficiently expressive to approximate any continuous invariant function. The proof we provide only assume that the group is compact and that the representation is real or complex, making it applicable to a wide range of problems. We structure the rest of the paper as follows. In Section 2, we provide the necessary background on group actions, invariant polynomials, and separating sets. In Section 3, we introduce polarization methods and show how they can be used to construct separating sets of invariant polynomials, formulating several results regarding the generation of separating families of invariant polynomials. In Section 4, we discuss how polarization can be used to construct invariant architectures and what cases are most suitable for practical applications. Finally, in Section 5, we summarize our contributions and discuss future research directions.

## 2 INVARIANT POLYNOMIALS FOR FUNCTION APPROXIMATION

### 2.1 GROUP THEORY

#### 2.1.1 GROUP ACTIONS AND REPRESENTATIONS

Let $V$ be a vector space of finite dimension over the field $\mathbb{K}$ (either $\mathbb{R}$ or $\mathbb{C}$) and $G$ a compact group acting linearly on $V$ via the action $\rho_V$, which is a group homomorphism from $G$ to $GL(V)$, the group of invertible linear transformations on $V$. We say that $V$ is a representation of $G$, or a $G$-representation. $V$ is said to be irreducible if it does not contain any proper subspace that is stable under the action of $G$. According to Maschke's theorem, any representation of a finite group can be decomposed as a direct sum of irreducible representations, and this result is known to extend to compact groups as well. Additionally, for finite groups, there is only a finite number of irreducible representations up to isomorphism.

#### 2.1.2 INVARIANCE AND EQUIVARIANCE

For any function $f : V \mapsto \mathbb{K}$, we say that $f$ is invariant with respect to the action of $G$ if $f \circ \rho_V = f$, that is

$$f(\rho_V(g)(\boldsymbol{v})) = f(\boldsymbol{v}), \quad \forall g \in G, \forall \boldsymbol{v} \in V. \tag{1}$$

This concept can be generalized to the notion of equivariance. For a function $f : V \mapsto W$, where $W$ is another vector space with a linear action $\rho_W$ of $G$, we say that $f$ is equivariant with respect to the action of $G$ if $f \circ \rho_V = \rho_W \circ f$. The case of invariance can be seen as a special case of equivariance by considering the trivial action of $G$ on $\mathbb{K}$ defined by $\rho_{\mathbb{K}}(g)(\boldsymbol{\lambda}) = \boldsymbol{\lambda}$ for all $g \in G$ and $\boldsymbol{\lambda} \in \mathbb{K}$.

#### 2.1.3 THE RING OF INVARIANT POLYNOMIALS

Let $\mathbb{K}[V]$ be the symmetric algebra of $V^*$, the dual space of $V$. $\mathbb{K}[V]$ can be identified with the polynomials on $V$ by choosing a basis $(x_1, \ldots, x_d)$ of $V^*$ that form the indeterminates. We denote by $\mathbb{K}[V]^G$ the subalgebra of $\mathbb{K}[V]$ consisting of $G$-invariant polynomials. It is known from the finiteness theorem of Hilbert (1890) that $\mathbb{K}[V]^G$ is finitely generated as an algebra, i.e. there exists a finite set of invariant polynomials $p_1, \ldots, p_n$ such that any invariant polynomial $p \in \mathbb{K}[V]^G$ can be expressed as a polynomial in $p_1, \ldots, p_n$. In addition, $\mathbb{K}[V]^G$ can be obtained from $\mathbb{K}[V]$ via the Reynolds operator, which is the surjective map $\mathcal{R}_G : \mathbb{K}[V] \mapsto \mathbb{K}[V]^G$ defined by

$$\mathcal{R}_G(p) = \int_G p \circ \rho_V(g) d\mu(g), \tag{2}$$

where $\mu$ is the Haar measure on $G$.

#### 2.1.4 APPROXIMATION OF INVARIANT FUNCTIONS BY INVARIANT POLYNOMIALS

The Stone-Weierstrass theorem can be used to prove that invariant polynomials are dense in the space of continuous invariant functions with compact support, with respect to the supremum norm. Such a proof relies on the Reynolds operator to construct invariant polynomials that approximate any continuous invariant function. This procedure, when paired with Hilbert's finiteness theorem, provides a constructive way to approximate any continuous invariant function from a finite number of invariant polynomials.

For finite groups, the degree bounds given by Noether (1915) ensure that $\mathbb{K}[V]^G$ admits a generating set of invariant polynomials of degree at most $|G|$, which can be computed using the Reynolds operator applied to the monomials of degree up to $|G|$. This bound is known to be tight for certain group actions. Unfortunately, the search for generators of $\mathbb{K}[V]^G$ is known to be a computationally hard problem given the exponential scaling of the number of polynomials to search given by Noether's bound. For infinite groups, the situation can be more complex, and the degree of the generators can be arbitrarily large. For this reason, neural network architectures that enforce invariance or equivariance for infinite groups often rely on the known characterization of the set of invariants of the most common groups. For example, the action of the orthogonal group $O(m)$ on $\mathbb{R}^{m \times n}$ results in a ring of polynomials generated by the inner products of the columns of the input matrix (Weyl, 1946), which is the basis for multiple invariant architectures for point clouds (Qi et al., 2017; Li et al.,

2018; Zhang et al., 2019). An overview of existing computational methods for finding generators of $\mathbb{K}[V]^G$ can be found in Derksen & Kemper (2015).

### 2.1.5 SEPARATING SETS OF INVARIANT POLYNOMIALS

Recently, promising theoretical approaches on the universality properties of invariant and equivariant architectures have forgone the use of generating sets of polynomials and have instead considered separating sets of polynomials, which can be defined as follows. A set of invariant polynomials $\{p_1, \ldots, p_n\} \subset \mathbb{K}[V]^G$ is said to be a separating set if for any two vectors $\boldsymbol{v}_1, \boldsymbol{v}_2 \in V$ such that $p_i(\boldsymbol{v}_1) = p_i(\boldsymbol{v}_2)$ for all $i = 1, \ldots, n$, there exists a group element $g \in G$ such that $\boldsymbol{v}_2 = \rho_V(g)(\boldsymbol{v}_1)$. Separating sets are particularly important for universality results, as they can be used to construct invariant architectures that can approximate any continuous invariant function with compact support, as shown by Azizian & marc lelarge (2021). Separating sets are also known to be easier to compute than generating sets and to be smaller in size. In addition, the technique of polarization, which we introduce in the next section, can be used to construct separating sets of invariant polynomials on high-dimensional representation that can be represented as direct sums of smaller spaces on which a group acts diagonally.

### 2.1.6 A NOTE ON EQUIVARIANCE

The concept of invariant polynomials extends naturally to equivariant polynomials. For $V$ and $W$ two $G$-representations, the set of equivariant polynomials from $V$ to $W$, denoted by $\mathrm{Mor}[V, W]^G$, is defined as the set of polynomial maps $f : V \mapsto W$ such that $f \circ \rho_V = \rho_W \circ f$. It is known that $\mathrm{Mor}[V, W]^G$ is a finitely generated module over the ring $\mathbb{K}[V]^G$, meaning that there exists a finite set of equivariant polynomials $f_1, \ldots, f_n \in \mathrm{Mor}[V, W]^G$ such that any equivariant polynomial $f \in \mathrm{Mor}[V, W]^G$ can be expressed as

$$f = \sum_{i=1}^{n} p_i f_i, \tag{3}$$

where $p_i \in \mathbb{K}[V]^G$ for all $i = 1, \ldots, n$. For this reason, the construction of equivariant architectures is typically achieved by combining invariant nonlinear functions with known equivariant functions, such as linear equivariant layers or equivariant polynomials (nathaniel thomas et al., 2018; Anderson et al., 2019).

We note that the theoretical results that we apply to invariant polynomials in the next section can be extended to equivariant polynomials as well. To do so, it is enough to realize $\mathrm{Mor}[V, W]^G$ corresponds to the invariant polynomials with variables in $V \times W^*$ that have degree in $W^*$ equal to one. For the details of this construction, we refer the reader to Section 4.2.3 of Derksen & Kemper (2015).

## 3 POLARIZATION METHODS FOR SEPARATING SETS OF INVARIANT POLYNOMIALS

### 3.1 POLARIZATION

### 3.1.1 POLARIZATION DEFINITION

We now introduce the concept of polarization, as well as the closely related concepts of simple and cheap polarization. Keeping the notations $V$ and $W$ for vector spaces on which $G$ acts linearly, we denote by $V^n$ the direct sum of $n$ copies of $V$. Given integers $m$ and $n$, polarization produces polynomials in $\mathbb{K}[V^n \oplus W]$ from polynomials in $\mathbb{K}[V^m \oplus W]$ in the following way. Introducing the indeterminates $\{X_{ij} \mid 1 \leq i \leq m, 1 \leq j \leq n\}$, for every $(\boldsymbol{v}_1, \ldots, \boldsymbol{v}_n) \in V^n$ and $\boldsymbol{w} \in W$, we can write

$$f\left(\sum_{j=1}^{n} X_{1j}\boldsymbol{v}_j, \ldots, \sum_{j=1}^{n} X_{mj}\boldsymbol{v}_j, \boldsymbol{w}\right) = \sum_{\boldsymbol{\alpha} \in \mathbb{N}^{m \times n}} f_{\boldsymbol{\alpha}}(\boldsymbol{v}_1, \ldots, \boldsymbol{v}_n, \boldsymbol{w})\boldsymbol{X}^{\boldsymbol{\alpha}}, \tag{4}$$

where $\boldsymbol{\alpha} = (\alpha_{11}, \ldots, \alpha_{mn})$ is a multi-index, $\boldsymbol{X}^{\boldsymbol{\alpha}} = X_{11}^{\alpha_{11}} \ldots X_{mn}^{\alpha_{mn}}$ and $f_{\boldsymbol{\alpha}} \in \mathbb{K}[V^n \oplus W]$. For $\mathbb{S} \subset \mathbb{K}[V^m \oplus W]$, the *polarization* of $\mathbb{S}$ to $n$ copies of $V$ is defined as the set

$$\mathrm{pol}_n \mathbb{S} = \left\{ f_{\boldsymbol{\alpha}} \mid f \in \mathbb{S}, \boldsymbol{\alpha} \in \mathbb{N}^{n \times d} \right\} \subset \mathbb{K}[V^n \oplus W]. \tag{5}$$

Note that if $\mathbb{S}$ is finite, $\mathrm{pol}_n \, \mathbb{S}$ is also finite since there are only a finite number of non-zero $f_{\boldsymbol{\alpha}}$ polynomials for each $f \in \mathbb{S}$. If $f \in \mathbb{K}[V^m \oplus W]^G$, then clearly, $f_{\boldsymbol{\alpha}} \in \mathbb{K}[V^n \oplus W]^G$ for all $\boldsymbol{\alpha} \in \mathbb{N}^{n \times d}$, which implies that if $\mathbb{S} \subset \mathbb{K}[V^m \oplus W]^G$, $\mathrm{pol}_n \, \mathbb{S} \subset \mathbb{K}[V^n \oplus W]^G$. In addition, if $f$ is homogeneous, the degree of the non-zero $f_{\boldsymbol{\alpha}}$ polynomials is the same as the degree of $f$.

### 3.1.2 POLARIZATION FOR GENERATING AND SEPARATING SETS

By Weyl's theorem, if $m \geq \min(n, \dim(V))$, a generating set of $\mathbb{K}[V^n \oplus W]^G$ can be obtained by polarization from a generating set of $\mathbb{K}[V^m \oplus W]^G$ (Kraft & Procesi, 1996). Draisma et al. (2008) extended this result to separating sets, showing that under the same conditions, a separating set of $\mathbb{K}[V^n \oplus W]^G$ can be obtained by polarization from a separating set of $\mathbb{K}[V^m \oplus W]^G$.

### 3.1.3 POLARIZATION VARIANTS

For $m \geq \dim(V) + 1$, an even simpler procedure named *simple polarization* is given by Domokos (2007). For $\mathbb{S} \subset \mathbb{K}[V^m \oplus W]^G$, the simple polarization of $\mathbb{S}$ to $n$ copies of $V$, defined for $n \geq m$, is the set

$$\mathrm{pol}_n^{\mathrm{simple}} \, \mathbb{S} = \left\{ f \circ \pi_{(i_1,\ldots,i_m)} \mid f \in \mathbb{S}, 1 \leq i_1 < \ldots < i_m \leq n \right\} \subset \mathbb{K}[V^n \oplus W]^G, \qquad (6)$$

where $\pi_{(i_1,\ldots,i_m)} : V^n \oplus W \mapsto V^m \oplus W$ is the projection onto the components $i_1, \ldots, i_m$ of $V^n$. The author of simple polarization proved that if $\mathbb{S} \subset \mathbb{K}[V^m \oplus W]^G$ is a separating set, then $\mathrm{pol}_n^{\mathrm{simple}} \, \mathbb{S}$ is a separating set of $\mathbb{K}[V^n \oplus W]^G$.

Draisma et al. (2008) also introduced a variant of polarization called *cheap polarization*, which only relies on a single indeterminate $a$ and on polynomials of $\mathbb{K}[V \oplus W]$ (instead of $\mathbb{K}[V^m \oplus W]$). This procedure is defined similarly to polarization, by expanding homogeneous polynomials $f$ of degree $d$ into

$$f\left( \sum_{j=1}^{n} a^{j-1} \boldsymbol{v}_j, \boldsymbol{w} \right) = \sum_{k=0}^{d \cdot (n-1)} f_k(\boldsymbol{v}_1, \ldots, \boldsymbol{v}_n, \boldsymbol{w}) a^k, \qquad (7)$$

The cheap polarization of $\mathbb{S} \subset \mathbb{K}[V^m \oplus W]$ to $n$ copies of $V$ is defined as the set

$$\mathrm{pol}_n^{\mathrm{cheap}} \, \mathbb{S} = \{ f_k \mid f \in \mathbb{S}, 0 \leq k \leq d \cdot (n-1) \} \subset \mathbb{K}[V^n \oplus W]. \qquad (8)$$

Unlike polarization and simple polarization, cheap polarization requires that $G$ be a finite group for separation to carry over. In this case, it is enough to set $m = 1$, such that if $\mathbb{S}$ is a separating set of $\mathbb{K}[V \oplus W]^G$, then $\mathrm{pol}_n^{\mathrm{cheap}} \, \mathbb{S}$ is a separating set of $\mathbb{K}[V^n \oplus W]^G$.

The fact that all three polarization methods are defined with $W$ as an additional representation of $G$ allows us to recursively create separating sets of invariant polynomials on G-representations $V$, decomposed as

$$V = V_1^{n_1} \oplus V_2^{n_2} \oplus \cdots \oplus V_k^{n_k}, \qquad (9)$$

by successive polarizations from a separating set of invariant polynomials on $V_1^{m_1} \oplus V_2^{m_2} \oplus \cdots \oplus V_k^{m_k}$, where $m_i \geq \min(n_i, \dim(V_i))$ for polarization, $m_i \geq \dim(V_i) + 1$ for simple polarization, and $m_i \geq 1$ for cheap polarization. For $\mathbb{S} \subset \mathbb{K}[V_1^{m_1} \oplus V_2^{m_2} \oplus \cdots \oplus V_k^{m_k}]^G$, we therefore define

$$\mathrm{pol}_{n_1,n_2,\ldots,n_k} \, \mathbb{S} = \mathrm{pol}_{n_k} \ldots \mathrm{pol}_{n_2} \, \mathrm{pol}_{n_1} \, \mathbb{S}, \qquad (10)$$

where each $\mathrm{pol}_{n_i}$ operator is applied to the $V_i^{m_i}$ component of its input. $\mathrm{pol}_{n_1,n_2,\ldots,n_k}^{\mathrm{simple}} \, \mathbb{S}$ and $\mathrm{pol}_{n_1,n_2,\ldots,n_k}^{\mathrm{cheap}} \, \mathbb{S}$ are defined similarly.

### 3.1.4 LIMITATIONS AND PRACTICAL CONSIDERATIONS

Given the constructive nature of polarization, the set $\mathrm{pol}_n \, \mathbb{S}$ can be computed explicitly from $\mathbb{S}$ using equation 4. This is done by expanding each monomial of $f$ in the indeterminates $\{x_{ij}\}$ and collecting terms with equal powers of the $x_{ij}$'s, which requires computing $n^d$ terms for each monomial of degree $d$. It is clear that even for relatively low degrees of invariant polynomials, the number of terms to consider becomes quickly intractable as $n$ increases, even if the determination of $\mathrm{pol}_n \, \mathbb{S}$ for the purpose of constructing invariant architectures needs to be done only once. In the

case of $\text{pol}_n^{\text{simple}} \, \mathbb{S}$, the separating set is given explicitly, but still includes a relatively large number of polynomials, equal to $|\mathbb{S}| \binom{n}{m}$, which is asymptotically equivalent to $|\mathbb{S}| n^m$ for large $n$. Including $\text{pol}_n^{\text{simple}} \, \mathbb{S}$ in an invariant architecture would therefore require computing and storing a very large number of features and present a serious bottleneck for most practical application. The case of cheap polarization is much more manageable, as the number of polynomials in $\text{pol}_n^{\text{cheap}} \, \mathbb{S}$ scales linearly with $n$. The fact that cheap polarization is applicable to finite groups only also ensures that $d$ is bounded by $|G|$ when $\mathbb{S}$ is a generating set of $\mathbb{K}[V]^G$. However, the restriction to finite groups means that many problems of interest, such as the action of the orthogonal group on point clouds, cannot be addressed using cheap polarization.

On the bright side, the large number of polynomials produced by polarization and simple polarization does not mean that all these polynomials are necessary for separating orbits. In fact, cardinality bounds on separating sets suggest that both $\text{pol}_n \, \mathbb{S}$ and $\text{pol}_n^{\text{simple}} \, \mathbb{S}$ are excessively large for the single purpose of separation as $n$ grows. Indeed, if $\mathbb{K}$ is algebraically closed, a separating set of size at most $2\dim(V) + 1$ always exists for $\mathbb{K}[V]^G$, as proven by Dufresne (2008). This proof is constructive and therefore provides a method to obtain such a small separating set from any separating set. This result, which is directly applicable to $\mathbb{C}$, can also be adapted to $\mathbb{R}$ by considering the complexification of $V$ and $G$, leading to a higher bound on the size of a separating set. However, we note that methods to construct a small separating set from a given separating set are of limited utility if an original large set needs to be computed first.

### 3.2 Small separating sets from separating families of invariant polynomials

Recently, Dym & Gortler (2025) proved a result that can be conveniently combined with any polarization method to obtain small separating sets of invariant polynomials, without requiring the computation of a large separating set. While their result relies on a hypothesis of strong separability for families of semi-algebraic functions, it can be simplified to a weaker statement in the context of this paper. We first introduce the concept of a separating family of invariant polynomials.

**Definition 1.** Let $V$ be a $G$-representation over $\mathbb{R}$ and $p$ a positive integer. A family of functions $\varphi \in \mathbb{R}[V \times \mathbb{R}^p]$ is said to be separating if for any two vectors $\boldsymbol{v}_1, \boldsymbol{v}_2 \in V$ in different orbits, there exists $\boldsymbol{\lambda} \in \mathbb{R}^p$ such that $\varphi(\boldsymbol{v}_1, \boldsymbol{\lambda}) \neq f(\boldsymbol{v}_2, \boldsymbol{\lambda})$.

Using this definition, we can state the following theorem.

**Theorem 1.** *Let $V$ be a $G$-representation of dimension $d$ over $\mathbb{R}$ and $p$ a positive integer. If $\varphi \in \mathbb{R}[V \times \mathbb{R}^p]$ is a separating family of invariant polynomials, then there are $2d + 1$ parameters $\boldsymbol{\lambda}_1, \ldots, \boldsymbol{\lambda}_{2d+1} \in \mathbb{R}^p$ such that the set*

$$\mathbb{S} = \{\varphi(\cdot, \boldsymbol{\lambda}_i) \mid i = 1, \ldots, 2d+1\} \subset \mathbb{R}[V]^G \tag{11}$$

*is a separating set of $\mathbb{R}[V]^G$.*

The proof of this theorem is an elementary consequence of the Theorem 2.7 of Dym & Gortler (2025) and is given in Appendix 6.1.

### 3.3 Learned polarization for small separating sets

We now show how Theorem 1 can be combined with polarization methods to obtain small separating sets of invariant polynomials from an irreducible decomposition of $V$, given by

$$V = V_1^{n_1} \oplus V_2^{n_2} \oplus \cdots \oplus V_k^{n_k}. \tag{12}$$

We write $d = \dim(V) = \sum_{i=1}^k n_i \dim(V_i)$. For $\mathbb{S} \subset \mathbb{K}[V_1^{m_1} \oplus V_2^{m_2} \oplus \cdots \oplus V_k^{m_k}]^G$ finite, with $m_1, \ldots, m_k$ some integers, we denote by $s = |\mathbb{S}|$ the size of $\mathbb{S}$ and write $f_1, \ldots, f_s$ its elements. For $\boldsymbol{v} \in V$, we denote by $\boldsymbol{v}_i$ the component of $\boldsymbol{v}$ in $V_i^{n_i}$ and $\boldsymbol{v}_{ij}$ the $j$-th component of $\boldsymbol{v}_i$ in $V_i$.

#### 3.3.1 Standard polarization

We start with the case of standard polarization. Let $\varphi_{\mathbb{S}}$ be the map defined by

$$\varphi_{\mathbb{S}} : \begin{array}{ccc} V \times \mathbb{K}^s \times \mathbb{K}^{m_1 \times n_1} \times \cdots \times \mathbb{K}^{m_k \times n_k} & \to & \mathbb{K} \\ (\boldsymbol{v}, \boldsymbol{\lambda}, \boldsymbol{X}_1, \ldots, \boldsymbol{X}_k) & \mapsto & \sum_{j=1}^s \lambda_j f_j(\boldsymbol{X}_1 \boldsymbol{v}_1, \ldots, \boldsymbol{X}_k \boldsymbol{v}_k), \end{array} \tag{13}$$

where $\boldsymbol{X}_i \boldsymbol{v}_i$ is in $V_i^{m_i}$ and whose $j$-th component for $j = 1, \ldots, m_i$ is equal to $\sum_{l=1}^{n_i} X_{ijl} \boldsymbol{v}_{il} \in V_i$.

**Proposition 1.** *For every* $i = 1, \ldots, k$, *assume* $m_i \geq \min(n_i, \dim(V_i))$. *Assume* $\mathbb{S}$ *is a separating set. Then,* $\varphi_{\mathbb{S}}$ *defined by equation 13 is a separating family of invariant polynomials of* $\mathbb{K}[V]^G$.

The main idea behind the proof starts from equation 5. It consists in realizing that each individual polynomial $f_{\boldsymbol{\alpha}}$ on the right-hand side of this equation can be obtained by a linear combination of multiple $f$'s evaluations where the indeterminate $\boldsymbol{X}$ is substituted by appropriate values. This construction is a classical result in interpolation theory based on the use of a Vandermonde matrix. The proof is given in Appendix 6.2.

Using Proposition 1 and Theorem 1, we can now conclude that there are $2d + 1$ parameters $\mathrm{w}_i \in \mathbb{K}^s \times \mathbb{K}^{m_1 \times n_1} \times \cdots \times \mathbb{K}^{m_k \times n_k}$, $i = 1, \ldots, 2d + 1$, such that the set

$$\mathbb{S}' = \{\varphi_{\mathbb{S}}(\cdot, \mathrm{w}_i) \mid i = 1, \ldots, 2d + 1\} \subset \mathbb{K}[V]^G \tag{14}$$

is a separating set of $\mathbb{K}[V]^G$.

### 3.3.2 SIMPLE POLARIZATION

The case of simple polarization is very similar and relies on the same $\varphi_{\mathbb{S}}$.

**Proposition 2.** *For every* $i = 1, \ldots, k$, *assume* $m_i \geq \dim(V_i) + 1$. *If* $\mathbb{S}$ *is a separating set, then,* $\varphi_{\mathbb{S}}$ *defined by equation 13 is a separating family of invariant polynomials of* $\mathbb{K}[V]^G$.

Note that Proposition 1 implies 2, since $\dim V_i + 1 > \min(n_1, \dim V)$. However, we provide a simple proof that does not rely on Proposition 1.

*Proof.* It is easy to see that the parameters $\boldsymbol{X}_i$ can be chosen such that $\boldsymbol{X}_i \boldsymbol{v}_i$ is equal to any projection of $\boldsymbol{v}_i$ onto $m_i$ components. Then, setting $\boldsymbol{\lambda}$ as the vector with 1 at the $j$-th position and 0 elsewhere, we obtain that $\varphi_{\mathbb{S}}(\cdot, \boldsymbol{\lambda}, \boldsymbol{X}_1, \ldots, \boldsymbol{X}_k)$ is equal to $f_j$ evaluated at any desired projection of $\boldsymbol{v}$ onto $V_1^{m_1} \oplus V_2^{m_2} \oplus \cdots \oplus V_k^{m_k}$. Therefore, $\mathrm{pol}_{n_1, n_2, \ldots, n_k}^{\mathrm{simple}} \mathbb{S} \subset \{\varphi_{\mathbb{S}}(\cdot, \mathrm{w}) \mid \mathrm{w} \in \mathbb{K}^s \times \mathbb{K}^{m_1 \times n_1} \times \cdots \times \mathbb{K}^{m_k \times n_k}\}$. Since $\mathbb{S}$ is separating, $\mathrm{pol}_{n_1, n_2, \ldots, n_k}^{\mathrm{simple}} \mathbb{S}$ is separating, and therefore $\varphi_{\mathbb{S}}$ is a separating family. $\square$

### 3.3.3 CHEAP POLARIZATION

For cheap polarization, the number of parameters involved in the definition of $\varphi$ is significantly reduced, given that only one indeterminate is used in equation 7. We make the additional assumption that $m_i = 1$ for all $i = 1, \ldots, k$. $\varphi_{\mathbb{S}}^{\mathrm{cheap}}$ is then defined as

$$\varphi_{\mathbb{S}}^{\mathrm{cheap}} : \quad \begin{array}{ccc} V \times \mathbb{K}^s \times \mathbb{K} & \to & \mathbb{K} \\ (\boldsymbol{v}, \boldsymbol{\lambda}, a) & \mapsto & \sum_{j=1}^s \lambda_j f_j \left( \sum_{l=1}^{n_1} a_j^{l-1} \boldsymbol{v}_{1l}, \ldots, \sum_{l=1}^{n_k} a_j^{l-1} \boldsymbol{v}_{kl} \right). \end{array} \tag{15}$$

**Proposition 3.** *For every* $i = 1, \ldots, k$, *assume* $m_i = 1$ *and that* $G$ *is a finite group. If* $\mathbb{S}$ *is a separating set, then,* $\varphi_{\mathbb{S}}^{\mathrm{cheap}}$ *defined by equation 15 is a separating family of invariant polynomials of* $\mathbb{K}[V]^G$.

The proof of this proposition, which is similar to the proof of Proposition 1, is given in Appendix 6.3

## 4 DISCUSSION ON APPLICABILITY AND PRACTICAL USE

The results presented in this paper provide a theoretical framework to construct small separating sets of invariant polynomials from an irreducible decomposition of a $G$-representation $V$. In particular, they are particularly interesting when $V$ is a high-dimensional representation that can be decomposed as $V = V_1^{n_1} \oplus V_2^{n_2} \oplus \cdots \oplus V_k^{n_k}$, In this case, the polarization methods presented in this paper can be used to construct separating sets of invariant polynomials on $V$ from separating sets of invariant polynomials on $V_1^{m_1} \oplus V_2^{m_2} \oplus \cdots \oplus V_k^{m_k}$, under the conditions on the $m_i$'s given in Propositions 1, 2 and 3. Therefore, the problem of constructing a separating set of invariant polynomials on $V$ is reduced to the problem of constructing a separating set of invariant polynomials on a lower-dimensional representation, making it more tractable to compute such a separating set for

the purpose of building invariant architectures. In addition, the number of parameters involved in polarization as well as the number of polynomial evaluations both scale linearly with the dimension of $V$.

For high-dimensional representations of finite groups, the finite number of irreducible representations ensures that $k$ and $\dim(V_i)$ are bounded, making it the most compelling case for polarization. In addition, cheap polarization becomes applicable, which further reduces the number of parameters involved in the construction of small separating sets and well as the dimension of the space for which a set of separating invariants has to be determined. We also note that the powers of $a$ in equation 15 might look like a limitation because of the numerical instabilities that very large or small values might create. However, a powerful workaround is to constrain $a$ to be on the unit circle in $\mathbb{C}$, which ensures that all powers of $a$ have modulus equal to one. The identity theorem for holomorphic functions can then be used to show that the separation property still holds, since a non-zero polynomial can only have a finite number of roots on the unit circle. Problems on real vector spaces can also be addressed by considering the complexification of $V$.

A potential limitation of the approach presented in this paper is that, by being restricted to the diagonal action of a group on multiple copies of a base space, it cannot include additional group actions that permute these copies. We therefore do not expect this method to be the most appropriate for problems where the group action includes such permutations, such as in the case of learning on permutation-invariant sets of vectors. However, many problems of interest can be formulated where each copy of a base space is distinguishable from the others, such as in the case of learning on sequences of geometric objects, where each object in the sequence is associated with a position in time.

## 5 CONCLUSION

In this work, we have presented a novel theoretical framework for constructing invariant and equivariant neural network architectures by leveraging the mathematical tool of polarization. We have shown that for high-dimensional group representations that can be expressed as a direct sum of smaller representations, the challenging problem of finding a basis of invariant polynomials can be addressed by computing separating invariants from a lower-dimensional space. Our key contribution was to combine the method of polarization with recent results on the existence of small separating sets. By re-framing polarization as a parameterized, learnable operation, we proved that a compact set of learned invariant features is guaranteed to form a separating set, which is sufficient for universal function approximation.

This approach provides a constructive method for designing expressive invariant features, which can be integrated into neural network architectures to ensure the capacity to distinguish between different group orbits and therefore to approximate any continuous invariant function on the representation space. This framework transforms the task of analytically deriving a complete set of generators into a practical learning problem on a much lower-dimensional space. The method is particularly compelling for finite groups, where "cheap polarization" offers a highly efficient implementation.

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

## 6 APPENDIX

### 6.1 PROOF OF THEOREM 1

*Proof.* Theorem 1 is a direct consequence of Theorem 2.7 of Dym & Gortler (2025). In that theorem, the authors consider a family of semi-algebraic functions $\varphi \in \mathbb{R}[V \times \mathbb{R}^p]$ that is strongly separating, that is, such that for every $x, y \in V$ in different orbits, the set

$$\{w \in \mathbb{R}^p \mid \varphi(x, w) \neq \varphi(y, w)\} \tag{16}$$

has dimensions $\leq p - 1$. Under these conditions, they prove that for Lebesgue almost $2d + 1$ parameters $\boldsymbol{\lambda}_1, \ldots, \boldsymbol{\lambda}_{2d+1} \in \mathbb{R}^p$, the set $\mathbb{S} = \{\varphi(\cdot, \boldsymbol{\lambda}_i) \mid i = 1, \ldots, 2d + 1\}$ is a separating set of $\mathbb{R}[v]^g$. They also remark that families of polynomials functions are semi-algebraic and that are separating if and only if they are strongly separating. Therefore, if $\varphi$ is a separating family of invariant polynomials, it satisfies the hypotheses of Dym & Gortler (2025)'s theorem, such that there exist $2d + 1$ parameters $\boldsymbol{\lambda}_1, \ldots, \boldsymbol{\lambda}_{2d+1} \in \mathbb{R}^p$ such that $\mathbb{S} = \{\varphi(\cdot, \boldsymbol{\lambda}_i) \mid i = 1, \ldots, 2d + 1\}$ is a separating set of $\mathbb{R}[v]^g$. □

### 6.2 PROOF OF PROPOSITION 1

*Proof.* To prove that $\varphi$ as defined in equation 13 is a separating family, we start by showing that given $f \in \mathbb{K}[v^m \oplus w]$, for every $\boldsymbol{v} = (\boldsymbol{v}_1, \ldots, \boldsymbol{v}_n) \in v^n$, $\boldsymbol{w} \in w$,

$$\text{span}\left\{ f\left(\sum_{j=1}^n X_{1j}\boldsymbol{v}_j, \ldots, \sum_{j=1}^n X_{mj}\boldsymbol{v}_j, \boldsymbol{w}\right) \mid \boldsymbol{X} \in \mathbb{K}^{m \times n} \right\}$$

$$= \text{span}\left\{ f_{\boldsymbol{\alpha}}(\boldsymbol{v}_1, \ldots, \boldsymbol{v}_n, \boldsymbol{w}) \mid f_{\boldsymbol{\alpha}} \in \text{pol}_n\{f\} \right\}.$$

Since by definition of $\text{pol}_n\{f\}$,

$$f\left(\sum_{j=1}^n X_{1j}\boldsymbol{v}_j, \ldots, \sum_{j=1}^n X_{mj}\boldsymbol{v}_j, \boldsymbol{w}\right) = \sum_{\boldsymbol{\alpha} \in \mathbb{N}^{m \times n}} f_{\boldsymbol{\alpha}}(\boldsymbol{v}_1, \ldots, \boldsymbol{v}_n, \boldsymbol{w}) \boldsymbol{X}^{\boldsymbol{\alpha}},$$

it is clear that the $\subset$ inclusion holds. To prove the $\supset$ inclusion, we first define some $\beta \in \mathbb{N}^{m \times n}$ such that $\alpha_{ij} \leq \beta_{ij}$ for all $i \in \{1, \ldots, m\}$, $j \in \{1, \ldots, n\}$, and $\boldsymbol{\alpha}$ with $f_{\boldsymbol{\alpha}} \neq 0$. Then, for $i \in \{1, \ldots, m\}$, $j \in \{1, \ldots, n\}$, and $\boldsymbol{X}_{ij} \in \mathbb{K}^{\beta_{ij}+1}$, we define the Vandermonde matrix $\boldsymbol{V}_{ij} \in \mathbb{K}^{(\beta_{ij}+1) \times (\beta_{ij}+1)}$ such that

$$\boldsymbol{V}_{ij} = \begin{bmatrix} 1 & X_{ij,1} & X_{ij,1}^2 & \cdots & X_{ij,1}^{\beta_{ij}} \\ 1 & X_{ij,2} & X_{ij,2}^2 & \cdots & X_{ij,2}^{\beta_{ij}} \\ \vdots & \vdots & \vdots & & \vdots \\ 1 & X_{ij,\beta_{ij}+1} & X_{ij,\beta_{ij}+1}^2 & \cdots & X_{ij,\beta_{ij}+1}^{\beta_{ij}} \end{bmatrix}.$$

We also write $\boldsymbol{Y}_{\boldsymbol{\alpha}} \in \mathbb{K}^{m \times n}$ such that $Y_{\boldsymbol{\alpha},ij} = X_{ij,\alpha_{11}} \ldots X_{ij,\alpha_{mn}}$ for $i \in \{1, \ldots, m\}$, $j \in \{1, \ldots, n\}$, with $\boldsymbol{\alpha} \in \mathbb{N}^{m \times n}$, and $\tilde{f}_{\boldsymbol{\alpha}} = f\left(\sum_{j=1}^n Y_{\boldsymbol{\alpha},1j}\boldsymbol{v}_j, \ldots, \sum_{j=1}^n Y_{\boldsymbol{\alpha},mj}\boldsymbol{v}_j, \boldsymbol{w}\right)$. Then,

$$\boldsymbol{V}_{11} \otimes \boldsymbol{V}_{12} \otimes \cdots \otimes \boldsymbol{V}_{mn} \begin{bmatrix} f_{0,\ldots,0}(\boldsymbol{v}_1, \ldots, \boldsymbol{v}_n, \boldsymbol{w}) \\ f_{1,0,\ldots,0}(\boldsymbol{v}_1, \ldots, \boldsymbol{v}_n, \boldsymbol{w}) \\ \vdots \\ f_{\beta_{11},\ldots,\beta_{mn}}(\boldsymbol{v}_1, \ldots, \boldsymbol{v}_n, \boldsymbol{w}) \end{bmatrix} = \begin{bmatrix} \tilde{f}_{0,\ldots,0} \\ \tilde{f}_{1,0,\ldots,0} \\ \vdots \\ \tilde{f}_{\beta_{11},\ldots,\beta_{mn}} \end{bmatrix}. \tag{17}$$

It is a well-known property of Vandermonde matrices that they are invertible if the $X_{ij,k}$'s are all distinct. Therefore, it is possible to choose every $\boldsymbol{V}_{ij}$ invertible, such that the matrix $\boldsymbol{V}_{11} \otimes \boldsymbol{V}_{12} \otimes \cdots \otimes \boldsymbol{V}_{mn}$ is also invertible. Then, we can express each $f_{\boldsymbol{\alpha}}(\boldsymbol{v}_1, \ldots, \boldsymbol{v}_n, \boldsymbol{w})$ as a linear combination of the $\tilde{f}_i$'s, which proves the $\supset$ inclusion.

Successive applications of this result to each $f_j \in \mathbb{S}$ and decomposition of a space $V_1^{n_1} \oplus V_2^{n_2} \oplus \cdots \oplus V_k^{n_k}$ as in equation 12 lead to the fact than any polynomial in $\text{pol}_{n_1,n_2,\ldots,n_k} \mathbb{S}$ can be expressed as a linear combination of terms from the left-hand side of equation 4 evaluated at appropriate $\boldsymbol{X}$'s.

It remains now to prove that $\varphi_{\mathbb{S}}$ is separating. For two vectors $\boldsymbol{u}, \boldsymbol{v} \in V_1^{n_1} \oplus V_2^{n_2} \oplus \cdots \oplus V_k^{n_k}$ in different orbits, since $\mathbb{S}$ is separating, there exists $f_{\boldsymbol{\alpha}} \in \mathbb{S}$ such that

$$f_{\boldsymbol{\alpha}}(\boldsymbol{u}) \neq f_{\boldsymbol{\alpha}}(\boldsymbol{v}).$$

Writing $f_{\boldsymbol{\alpha}}(\boldsymbol{u})$ as a linear combination of terms of the form $f(\boldsymbol{X}_1 \boldsymbol{u}_1, \ldots, \boldsymbol{X}_k \boldsymbol{u}_k)$ (where we use the same notation as in equation 13) with $\boldsymbol{X}$'s fixed during the Vandermonde matrix construction, and doing the same for $f_{\boldsymbol{\alpha}}(\boldsymbol{v})$, since these linear combinations do not depend on $\boldsymbol{u}$ or $\boldsymbol{u}$, there must be an $\boldsymbol{X}$ such that

$$f(\boldsymbol{X}_1 \boldsymbol{u}_1, \ldots, \boldsymbol{X}_k \boldsymbol{u}_k) \neq f(\boldsymbol{X}_1 \boldsymbol{v}_1, \ldots, \boldsymbol{X}_k \boldsymbol{v}_k).$$

Choosing $\boldsymbol{\lambda}$ as the vector with 1 at the $\boldsymbol{\alpha}$-th position and 0 elsewhere, we obtain the desired result.

$\square$

### 6.3 PROOF OF PROPOSITION 3

*Proof.* The proof carries over directly from the proof of Proposition 1, with the main difference being that a single Vandermonde matrix is used that shows that each $f_k$ in equation 7 can be expressed as a linear combination of evaluated polynomial terms. $\square$

