# OpenReview forum: "Invariant and equivariant architectures via learned polarization"
_ICLR.cc/2026/Conference — Submitted to ICLR 2026_

### Official Review · Reviewer_de7C · 2025-10-17

**Soundness:** 2
**Presentation:** 3
**Contribution:** 1
**Rating:** 2
**Confidence:** 4

**Summary:**

The paper discusses how to construct (separating sets of) invariant/equivariant polynomials for high-dimensional group representations, starting from low-dimensional ones. This is based on the polarization method from invariant theory.

**Strengths:**

- The paper is well written: it is concise, clear, and the mathematical ideas are presented in a friendly and pedagogical way, despite being fully rigorous.

- The idea of relying on polarization to construct invariant/equivariant models is original, and can be potentially built upon in the context of geometric deep learning.

**Weaknesses:**

The main limitation of the paper is that, for the most part, it is not targeted towards machine learning / neural networks, which, in my opinion, is fundamental for a machine learning venue such as ICLR. While generating and separating sets are briefly motivated from the perspective of constructing neural networks (lines 106-114), this relation is never elaborated further. Instead, all the results, hypotheses, and constructions, are written in a purely-algebraic formalism, without any connection to neural networks. I do not see how the constructions of separating sets can be applied to obtain actual neural architectures. Some specific issues are:

- The constructions discussed in the paper are recursive, i.e., they require a starting separating set. It is unclear when and to what extent a starting separating set is known for representations used in practical applications.

- The paper focuses on invariant polynomials, as traditional in algebraic invariant theory, and the constructions crucially rely on their algebraic nature. However, neural networks typically do not define polynomials, so it is unclear how to apply these constructions to them.

- The constructions relying on standard and simple polarization require assumptions on the dimensions of the representations, and on their multiplicities (see line 169, 201-202, Proposition 1 and 2).  Again, it is unclear whether these assumptions are satisfied for commonly-deployed representations. Note that this is not an issue for cheap polarization, since it does not require this type of assumption.

Moreover, the paper does not provide a single example, even outside of the machine learning world. Instead, I believe it would greatly benefit from examples, especially related to neural networks, and from expanding on the connection to deep learning. As a side note, the paper is less than 7 pages long, implying that the lack of examples and of connection to deep learning is not related to space constraints.

In conclusion, I unfortunately believe that the paper, despite proposing an original and appealing idea, does not deserve acceptance. I am, of course, open to discussion.

Minor:

-   Formatting issue: I believe that some (sub-) sections are too short (e.g., 3.1.1 and 3.1.2), which is inappropriate formatting, I believe.

- Definition 1 looks redundant to me, since it reintroduces the notion of separating set, which already appears as early as in Section 2.1.5. I understand that the notion in Section 2.1.5 is defined only for finite sets, while Definition 1 is for infinite/continuous families (indexed by $\mathbf{\lambda} \in \mathbb{R}^p$). However, this is a mere convention: one can directly define a general notion of separateness for arbitrary families of functions. Why not introducing the notion of separating set for arbitrary families from the beginning, and, where appropriate, require them to be finite?

- Typo on line 319: the sentence ends in a comma, instead of a period.

**Questions:**

I would be grateful if the authors could elucidate on the main weakness raised above, i.e., the relation to actual neural network architectures. For example, could you explain how these polynomial constructions apply to neural networks? Could you provide examples of the resulting separating sets, and of representations appearing in practice that satisfy the assumptions?

---

### Official Review · Reviewer_NZvJ · 2025-11-01

**Soundness:** 4
**Presentation:** 4
**Contribution:** 1
**Rating:** 2
**Confidence:** 4

**Summary:**

The paper presents a principled method for constructing computationally efficient, separating, polynomial equivariant features.
They show that this enables the efficient construction of models universal in the class of continuous equivariant functions.
The approach repeatedly applies polarization (and variants) on the isotypic decomposition of the input representation to build such features.
The authors show that polarization and simple polarization are asymptotically equivalent in computational cost, and often intractable, whereas cheap polarization, though limited to finite groups, is typically more efficient.

**Strengths:**

- A clear and accurate exposition of the introductory material on classical invariant theory, polarization, and separating sets.
- The problem and research direction are relevant to the equivariant ML community, with both practical and theoretical potential.

**Weaknesses:**

- **Limited novelty and contribution.** The main propositions and theorems are direct applications of standard results. In particular, Theorem 1 is an immediate corollary of Theorem 1.7 in [1] (mis-cited in the manuscript as Theorem 2.7), while Propositions 1–3 follow directly from interpolation via the Vandermonde matrix, a standard technique in classical invariant theory (see, e.g., [2]).
- **Literature positioning is incomplete.** The paper should more thoroughly address related work in equivariant ML based on classical invariant theory and clearly position itself relative to fundamental work in the field such as [3, 4].
  To better address the graph-learning community, the authors might also cite [5], which studies equivariant polynomials as equivariant features in graph learning.
- **No empirical validation.** While empirical validation is not strictly required for this type of theoretical work, providing it where feasible would strengthen the contribution (see also Q1).
- **No illustrative examples.** Including simple, worked examples in concrete settings (e.g., the symmetric or cyclic groups) would substantially improve readability.

##### References:

[1] N. Dym and S. J. Gortler, *Low-dimensional invariant embeddings for universal geometric learning*, 2025 \
[2] J. Draisma et al., *Polarization of Separating Invariants*, 2008 \
[3] B. Blum-Smith and S. Villar, *Machine learning and invariant theory*, 2023 \
[4] B. Blum-Smith et al., *A Galois theorem for machine learning: Functions on symmetric matrices and point clouds via lightweight invariant features*, 2025 \
[5] O. Puny et al., *Equivariant Polynomials for Graph Neural Networks*, 2023

**Questions:**

1. Is it feasible to compute separating sets with softwere tools such as Magma, SageMath, or Macaulay2? More broadly, how should the proposed pipelines be implemented in practical ML workflows?
2. For groups where cheap polarization is unavailable, what concrete advantages polarization can offer over existing invariant bases?

---

### Official Review · Reviewer_aMZ2 · 2025-11-01

**Soundness:** 1
**Presentation:** 2
**Contribution:** 2
**Rating:** 2
**Confidence:** 4

**Summary:**

This paper proposes a novel theoretical framework for constructing invariant and equivariant neural networks that respect group symmetries. Whereas traditional approaches often rely on the "generating set" of the invariant ring, which is frequently computationally intractable, this work focuses on the less restrictive concept of a "separating set."

**Strengths:**

**Importance of the Problem Setting and Viewpoint:** The paper's premise—that identifying the "generating set of the invariant ring," which serves as the theoretical foundation for designing symmetric neural networks, is extremely difficult—is accurate and highly relevant.

**Introduction of "Separating Sets":** In response to this difficult problem, the theoretical approach of introducing the concept of a "separating set"—a weaker condition than requiring the complete information of the invariant ring (the generating set)—and attempting to construct an architecture that can universally approximate invariant functions using this set, is original.

**Weaknesses:**

**Complete Lack of Experimental Validation :** The paper remains a purely theoretical proposal, and no experiments whatsoever have been conducted to demonstrate the effectiveness, practicality, or limitations of the proposed framework. The fact that the proposed method theoretically guarantees the *existence* of a separating set is an entirely different matter from whether it can be stably *learned* as a machine learning model on actual data (e.g., via gradient descent) and whether it possesses practical advantages (e.g., data efficiency, generalization, computational cost) compared to existing methods. To claim theoretical soundness, minimal proof-of-concept (PoC) experiments (e.g., a demonstration on a simple finite group) are essential.

**Lack of Awareness of Related Work (Especially Practical Invariant/Equivariant Networks) :** The authors repeatedly claim that "existing methods require explicit knowledge of the generating set of the invariant ring," but this does not accurately reflect recent research trends. For example, many studies, led by Deep Sets (Zaheer et al., 2017) and Invariant Graph Networks (Maron et al., 2018), achieve permutation invariance (a type of symmetry) using simple operations like sum-pooling or averaging, without any complete knowledge of the invariant ring, and have demonstrated high practical utility. The theoretical contribution of this paper needs to be clearly discussed and compared with these (already practical) approaches to clarify its relationship and potential advantages. However, this contextualization is severely lacking, making it difficult to judge the paper's novelty and contribution.

**Questions:**

**Regarding the lack of experiments:** Why does this paper not include even minimal experiments (e.g., a proof-of-concept on a simple dataset with known symmetries) to validate the effectiveness of the proposed method? A theoretical "existence proof" does not necessarily guarantee practical effectiveness as a machine learning architecture in terms of learning stability, expressive power, or computational cost, does it?

**Regarding comparison to existing practical methods:** What are the theoretical and practical differences between the "learned polarization" proposed in this paper and existing methods that achieve invariance/equivariance without explicit generating sets of the invariant ring (e.g., sum-pooling), as exemplified by Maron et al. (2018) and Zaheer et al. (2017)?

---

### Official Review · Reviewer_hpb3 · 2025-11-03

**Soundness:** 3
**Presentation:** 2
**Contribution:** 1
**Rating:** 2
**Confidence:** 2

**Summary:**

This paper presents tools based on polarization from invariant theory and suggest that they can be used to build equivariant neural networks. Three variants of polarization are analyzed and a scheme for learned polarization is suggested.

**Strengths:**

- The paper presents interesting potential for bridging concepts from classical invariant theory in machine learning in a novel way

**Weaknesses:**

- I have to say that despite being quite familiar with equivariant machine learning most of the concepts developed in this paper were quite obscure to me. I think the writing is heavy and assume knowledge of algebraic invariant theory which is uncommon
- I think a very small audience within ICLR will be able to understand the paper and interest in it. I suggest instead submitting to an applied math venue, where the work will benefit from a more appropriate audience and better feedback
- The lack of proposal for a practical implementation or suggestion of specific application is also something that makes ICLR a suboptimal venue for this work

**Questions:**

-

---

### Meta-Review · Area_Chair_TRhR · 2025-12-18

**Summary:**

This paper proposes a theoretical framework for constructing invariant and equivariant neural network architectures using polarization methods from classical invariant theory. The paper was reviewed by four experts with strong backgrounds in equivariant machine learning and algebraic methods in deep learning. While all reviewers acknowledged that the paper presents an original idea with potential relevance to the geometric deep learning community, they unanimously raised significant concerns. The primary criticisms center on: (1) the lack of any experimental validation or proof-of-concept demonstrations; (2) insufficient connection to practical neural network architectures—the purely algebraic formalism leaves unclear how these polynomial constructions would translate to actual implementable models; (3) limited novelty, as some key results appear to be direct applications of standard techniques from classical invariant theory;  and (4) incomplete positioning relative to existing practical equivariant methods that achieve symmetry without explicit invariant ring characterization (DeepSets,IGNs).  The absence of illustrative examples, even simple ones for symmetric or cyclic groups, further limits accessibility and impact.

The consensus is that while the direction is interesting, the paper in its current form is better suited for an applied mathematics venue and does not meet the standards for acceptance at ICLR.

**Reviewer Concerns:**

no rebuttal

**Reviewer Scores:**

no rebuttal

---

### Decision · Program_Chairs · 2026-01-26

Reject